# An Emerging Solution for Medical Waste: Reuse of COVID-19 Protective Suit in Concrete

**Tao Ran [1], Jianyong Pang [1] and Jiuqun Zou [1,2,\***

1   School of Civil Engineering and Architecture, Anhui University of Science and Technology, Huainan 232001, China
2   China MCC 17 Group Co., Ltd., Maanshan 243061, China
\*   Correspondence: 2020136@aust.edu.cn

**Abstract:** With the continuous spread of COVID-19 (coronavirus disease 2019), a large number of medical protective suits (PS) have been used and discarded, causing great damage to the ecological environment. The main component of PS is polypropylene plastic, which will enter the oceans, rivers, and animals with groundwater and will not decompose for hundreds of years. Therefore, this global health crisis not only affects the health and economy of the world's population now but will also continue to disrupt our daily lives after the pandemic ends. The main objective of this study is to explore an effective method to reduce the biological and environmental hazards of medical waste by combining PS with concrete. Due to the excessive size of the PS, protective suit fibers (PSF) were obtained from PS by cutting. To investigate the possibility of using PS in concrete, a series of experiments were conducted, including a physical parameter test, compression test, split tensile test, ultrasonic pulse velocity test, scanning electron microscope (SEM), and finite element simulation. The results indicated that the introduction of PSF significantly enhanced the mechanical properties of concrete, and the maximum compressive strength and splitting tensile strength increased by 7.3% and 43.6%, respectively. The ultrasonic pulse velocity and density of concrete containing PSF decreased compared with the control group. The images of SEM show that PSF binds tightly to the cement matrix and hinders the propagation of micro-cracks. The introduction of PS into the concrete material leads to the improvement of the mechanical properties of concrete and the improvement of the overall quality of the concrete, which is of great significance for reducing the damage of medical waste to the environment. The originality of this work is that polypropylene fibers acquired from PS were put into concrete for the first time for performance testing.

**Keywords:** COVID-19; protective suit fibers; polypropylene plastic; concrete; finite element simulation

## 1. Introduction

In recent years, coronavirus disease 2019 (COVID-19) has become a major public health event, which has affected 223 countries with around 500 million confirmed infected cases and 6 million deaths until 2022 and this number is still increasing [1]. COVID-19 is a highly contagious respiratory infectious illness distributed via aerosols, big droplets (e.g., cough and sneeze), and direct contact with secretions or fomites [2]. Due to the ongoing pandemic, plastic-based personal protective equipment (PPE) for frontline health workers, as well as face masks for the general public, have been used to combat the spread of COVID-19 [3]. Face shields, surgical masks, protective suits, N95 respirators, and surgical gloves are examples of standard PPE [4]. During such an outbreak, a significant amount of PPE was produced [5]. Every month, it is estimated that 129 billion masks and 65 billion gloves were used over the world [6]. In February 2020, China manufactured 116 million PPE per day, a 12-fold increase from before the COVID-19 outbreak [7]. However, the widespread use of PPE will bring serious environmental consequences. Such materials are not easily biodegradable and may persist in the air, soil, or sea, posing significant risks to human

and animal health [8]. As a result, the global pandemic has had a significant impact on the environment that will continue to disrupt our daily lives even after the pandemic is over [9].

Disposable personal protective suits (PS) are a crucial part of PPE, with a higher volume and quality than general PPE and high recycling value. Nonwoven fabric is commonly used in the fabrication of PS because it allows for very quick and inexpensive manufacturing and this type of nonwoven fabric is typically made from polypropylene and ends up in a spunbond-meltblown-spunbond (SMS) construction [10]. The production process of synthetic fibers causes a large number of carbon emissions, for example, accounting for two-thirds of the total 10% of worldwide carbon emissions linked with textile products [11]. At the same time, the construction industry is one of the greatest consumers of resources and raw materials [12]. According to Worldwatch Institute data, building construction consumes 40% of the world's stone, sand, and gravel, 25% of the world's timber, and 16% of the world's water each year [13]. The production and transportation of building materials, as well as the installation and construction of buildings, consumed a lot of energy and released a large number of greenhouse gases (GHG) [12]. It may be able to reduce carbon emissions while also cutting building costs if PS is recycled in the construction sector, which is both environmentally and economically beneficial.

Polypropylene and plastic fiber have been widely concerned in civil engineering. Liang [14] explored the application of multi-scale polypropylene fiber (PPF) hybridization in roller-compacted concrete (RCC), the tensile stress–strain curves and corresponding tensile parameters of polypropylene fiber-reinforced roller-compacted concrete (PFRCC) are obtained. The results show that the strength and toughness of PFRCC have been significantly improved. Behfarnia and Behravan [15] studied the application of high-performance polypropylene fibers (HPPF) in water tunnel concrete linings, and the results show that HPPF has a higher effect on concrete flexural toughness, concrete permeability, and chloride ion permeability than steel fibers. Zia [16] investigated the cracking control impact of polypropylene fibers in canal lining concluding that polypropylene fibers can significantly slow the cracking speed of canal lining. Hussain [17] applied polypropylene fiber to the concrete pavement and found that polypropylene fiber-reinforced concrete has higher economic benefits than traditional ordinary concrete. Alani and Beckett [18] performed punching failure experiments with high-performance polypropylene fibers added to a ground support plate of 6.00 m × 6.00 m × 0.15 m, and the findings obtained were comparable to the punching experiment failure values of steel fiber plates under similar conditions. Khan [19] applied wave polypropylene fiber to concrete roads and found that the compressive strength, flexural strength, and split tensile strength of wave polypropylene fiber-reinforced concrete (WPFRC) were increased by 11.7%, 21.5%, and 17.5%, respectively, and the use of polypropylene wave fiber cost savings of 1.7% (per lane per kilometer).

The application of recycled plastic fibers in concrete has been reported. Bhogayata and Arora [20] mixed waste plastic fibers into concrete and found that concrete composites containing fibers showed considerable improvements in impact resistance and energy absorption compared to standard concrete. Foti [21] researched the use of polyethylene terephthalate (PET) fibers recovered from pet bottles in concrete, which is regarded as an important waste disposal method for the environment. Fraternali [22] investigated the effect of recycled PET fibers on the mechanical properties and seawater erosion of Portland cement-based concrete. Won [23] conducted research on the performance of recycled PET fiber-reinforced concrete in alkaline, acidic, and salt solution environments, showing that the durability of concrete incorporating recycled PET fibers is significantly enhanced. Bui [24] studied the effect of plastic fibers recovered from the woven plastic sacks on the performance of concrete and concluded that 0.25–0.75% of plastic fibers can improve concrete performance.

The fabric of PS used to improve the performance of concrete is non-woven, which is formed of polypropylene fibers and is regarded as a suitable modification material for con-

crete. At the same time, the PS used by residents and hospitals contributes to non-recyclable plastic trash and had a negative influence on the environment, as there was presently no infrastructure in place to handle possible pollution in a safe and environmentally friendly manner. It is critical to investigate how to dispose of these medical wastes in a responsible and safe manner.

The originality of this work is that polypropylene fibers acquired from PS were put into concrete for the first time for performance testing. To assess the feasibility of incorporating PS into concrete, a series of experiments were carried out, including physical parameter test, compressive test, splitting tensile test, ultrasonic pulse velocity (UPV) test, and scanning electron microscope (SEM) test. By merging discarded PS with concrete, this research may be able to propose new ideas for the recycling of PPE to lessen the ecological damage caused by the pandemic.

## 2. Materials and Methods

### 2.1. Materials

The P·C 42.5 composite Portland cement was used in this experiment, and the cementitious material standard conformed to the China General Portland Cement Standard GB 175-2007. The chemical composition and X-ray powder diffraction (XRD) results of the cement were provided in Table 1 and Figure 1. Ordinary river sand is used as fine aggregate. The coarse aggregate is crushed limestone with continuous particle gradation and particle sizes ranging from 5 to 20 mm, with an apparent density of 2700 kg/m$^3$. The particle size distribution curves of fine and coarse aggregates are given in Figure 2. The water-reducing admixture employs a liquid high-performance water-reducing from China Xi'an Building Materials Company with water-reducing effectiveness of 30% to assure the fluidity and water retention of concrete. The PS used in the experiment were produced by Yuan Company (Xinxiang, China). As the epidemic continues to spread, the use of contaminated PS may lead to the transmission of the virus in the laboratory and community. In order to reduce the risk of virus leakage, the PS used in the experiment is new and uncontaminated. The zippers and elastics on PS were removed before experiments, and PS were cut into pieces 2 cm long and 0.4 cm wide, a size that was mentioned in the previous study [25]. The specific shapes of the cut protective suit fibers (PSF) are listed in Figure 3a,b. The SEM testing of PSF observed a reticulated polypropylene fiber distribution and shown in Figure 3c, this spatial distribution may contribute to the mixing of PSF with concrete. In addition, some parameters of PSF are given in Table 2.

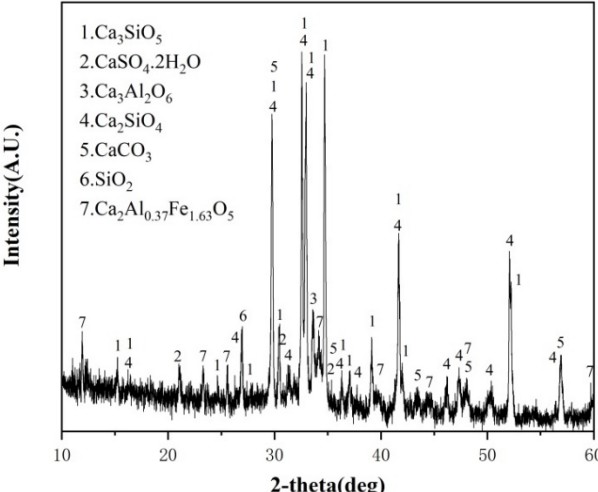

**Figure 1.** XRD pattern of cement.

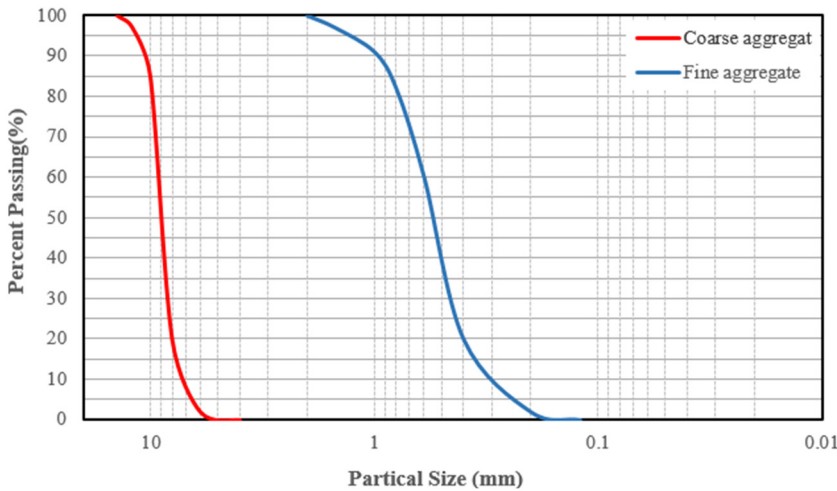

**Figure 2.** Particle size distribution curves of fine and coarse aggregates.

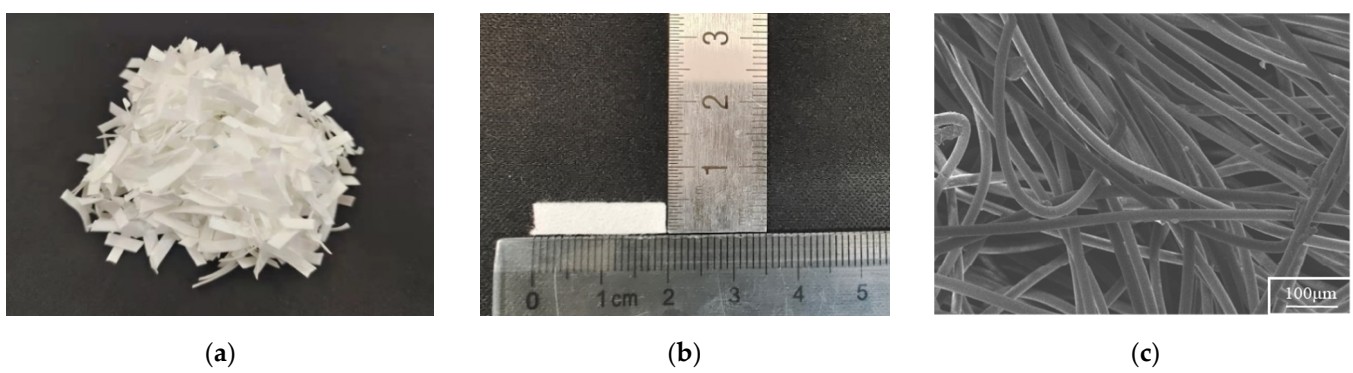

(**a**) (**b**) (**c**)

**Figure 3.** PSF specific size and SEM image: (**a**) PSF stacking morphology; (**b**) specific size; (**c**) PSF SEM image.

**Table 1.** The chemical compositions of cement.

| Composition Content (%) | Cement |
|---|---|
| $SiO_2$ | 22.60 |
| $Al_2O_3$ | 5.03 |
| $Fe_2O_3$ | 4.38 |
| CaO | 63.11 |
| MgO | 1.46 |
| $SO_3$ | 2.24 |
| Loss on Ignition | 1.18 |

**Table 2.** Properties of protective suit fibers (PSF).

| Fiber Properties | PSF |
|---|---|
| Breaking force (N) | 183 |
| Elongation at break (%) | 42 |
| Specific gravity | 0.93 |
| Water absorption | 7.6% |
| Aspect radio | 5 |

### 2.2. Mix Design, Casting Procedures, and Specimen-Making

The designed concrete water–cement ratio is 0.4, the proportion of cementitious material is 445 kg/m$^3$, and the designed compressive strength is 40 MPa. According to

the different amount of PSF content, six groups of different mix ratios were designed, and the volume replacement rates of PSF in concrete were 0, 0.2%, 0.4%, 0.6%, 0.8%, and 1.0%, respectively. Abdullah and El Aal [9] found that incorporating 0.5% HPPM (healthy personal protective materials) into soil enhanced soil performance while increasing the addition to 1% impaired performance. Therefore, based on previous research, this experiment refined the range of fiber content, focusing on the effect of 0–1% PSF content on concrete performance. Table 3 lists the specific concrete design ratio, PS0 means the control group without adding PSF, similarly PS6 means the content of PSF is 0.6%.

**Table 3.** Mixing proportions of the experiment (kg/m$^3$).

| Group Number | Cement (kg) | Sand (kg) | Limestone (kg) | Water (kg) | Water Reducer (kg) | PSF (% by Volume) |
|---|---|---|---|---|---|---|
| PS0 | 445 | 641 | 1139 | 178 | 3.9 | 0 |
| PS2 | 445 | 641 | 1139 | 178 | 3.9 | 0.2 |
| PS4 | 445 | 641 | 1139 | 178 | 3.9 | 0.4 |
| PS6 | 445 | 641 | 1139 | 178 | 3.9 | 0.6 |
| PS8 | 445 | 641 | 1139 | 178 | 3.9 | 0.8 |
| PS10 | 445 | 641 | 1139 | 178 | 3.9 | 1.0 |

It is required to follow the prescribed stages for casting in order to acquire an approved concrete specimen. Firstly, placed the weighed aggregates and PSF into the horizontal concrete agitator and mixed for 3 min. Then, cement was added into agitator and mixed for 2 min. After this step, the water mixed with the superplasticizers was slowly added into the dry materials, and continued to mix for 3 min. This is to ensure that all materials can be fully contacted, allowing the concrete to blend together. The mixed concrete was poured into the cube molds 3 times while using a vibration table to vibrate and compact. This process allows the molds to be completely filled with concrete without voids. After a curing time of 24 h, the concrete samples were removed from the molds and placed in a curing room with a temperature of 20 ± 2 °C and a relative humidity more than 95% for 28 days for testing. A group of 6 cubic specimens needs to be cast and tested for compressive strength and splitting strength, respectively.

### 2.3. Testing Methods

#### 2.3.1. Strength Test

The loading instrument used for concrete compressive and splitting tensile strength is the WDW-1000 computer-controlled universal testing machine (see Figure 4). The maximum range of the loading machine is 1000 KN. The compressive and splitting tensile strengths of the 100 × 100 × 100 mm cubic specimens were measured, and three specimens were evaluated for each mix design to remove the probable experimental error. As per the standards, the rate of loading should be controlled at 0.5 MPa/s in the compressive strength test and 0.05 MPa/s in the splitting tensile strength test.

#### 2.3.2. Ultrasonic Pulse Velocity (UPV)

The ultrasonic pulse velocity method is considered to be a more potential nondestructive testing method for evaluating the quality and properties of concrete [26]. Before the strength tests of the specimens, the ultrasonic pulse velocity test was performed on the specimens using an ultrasonic non-destructive testing instrument produced by Beijing Ultrasonic Detection Technology Company (Beijing, China). The elastomer was smeared on the specimens, and the receiving device and the transmitting device were closely attached to the concrete surface to ensure that the receiving and transmitting devices were in the same line. The wave velocity was recorded until a stable waveform appeared on the detector. The ultrasonic frequency was set to 50 kHz, the sampling period was 0.4 μs, and the emission voltage was 500 V. This test method allows the relative quality and internal condition of concrete to be assessed without damaging it.

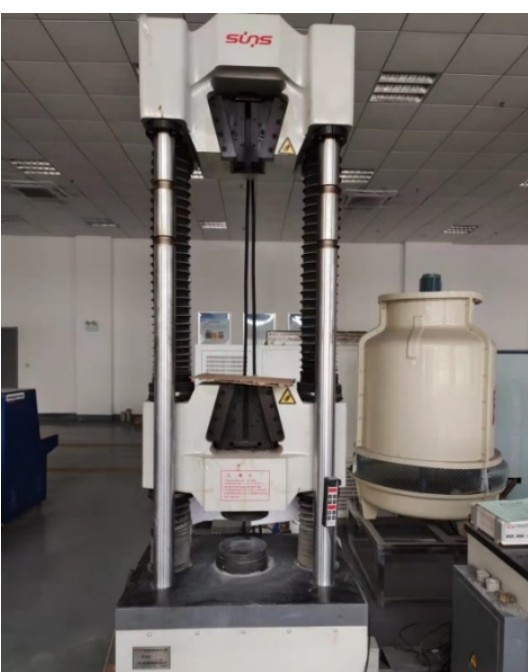

**Figure 4.** WDW-1000 Computer-Controlled Universal Testing Machine.

### 2.3.3. Scanning Electron Microscope Test

The microstructure of the samples was measured using the scanning electron microscope (FlexSEM1000). The sample dimensions were 5 mm wide and 2 mm thick, and the test pieces were taken from crushed concrete after the strength test. The role of PSF at the microscopic scale is understood by observing fiber distribution and contact in concrete.

In order to describe the experimental operation in detail, Figure 5 shows the specific flow of the experiment.

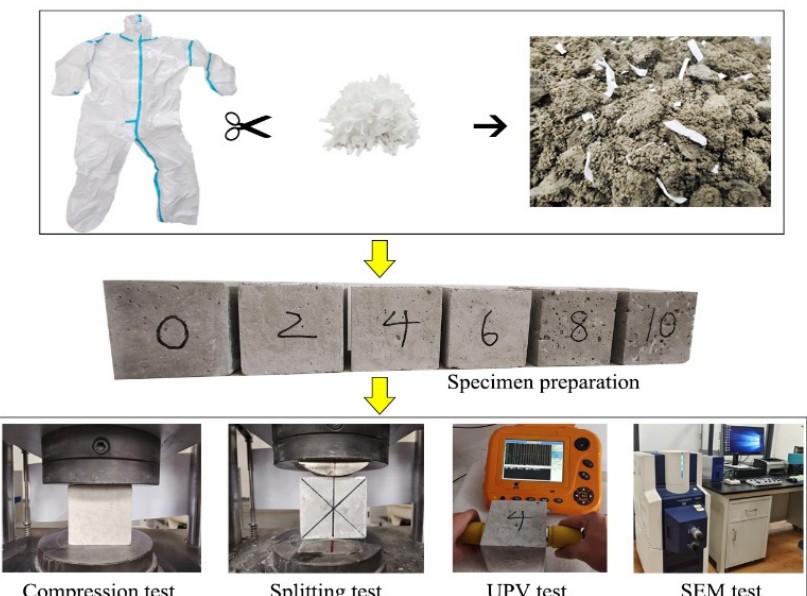

**Figure 5.** Specific operation process and experimental method. (the markers of 0, 2, 4, 6, 8, 10 means PSF content is 0, 0.2%, 0.4%, 0.6%, 0.8%, 1.0%, respectively).

## 3. Results and Discussion

### 3.1. Physical Parameters

The water–cement ratio, sand ratio, slump, and density of each group are given in Table 4. The water–cement ratio and sand ratio were consistent for each group, meaning their compositional compositions were very similar. However, for concrete with the same water–cement ratio and sand ratio, the slump exhibits a huge difference. The slumps of PS0, PS2, PS4, PS6, PS8, and PS10 are 34, 30, 28, 24, 22, and 17 mm, respectively. With the increase of PSF incorporation, the slump decreased gradually and the slump of PS2 and PS10 decreased by 11.7% and 50%, respectively, compared with PS0. It is possible that the connection and confinement of PSF reduced the fluidity of fresh concrete while increasing its consistency, resulting in a decrease in the slump of the experimental groups. This is the same as Qin's [27] experiment. Similarly, the density of concrete also decreased with the addition of PSF. The density of PS10 decreased by 1.6% compared with the control group. The inclusion of low-density PSF in the concrete to replace part of the denser concrete resulted in a lower density.

**Table 4.** Water–cement ratio, sand ratio, slump, and density of specimen.

| Group Number | Water–Cement Ratio W/C | SAND Ratio b1 (%) | Slump (mm) | Density (kg/m$^3$) |
|---|---|---|---|---|
| PS0 | 0.4 | 36.01 | 34 | 2402 |
| PS2 | 0.4 | 36.01 | 30 | 2399 |
| PS4 | 0.4 | 36.01 | 28 | 2392 |
| PS6 | 0.4 | 36.01 | 24 | 2383 |
| PS8 | 0.4 | 36.01 | 22 | 2377 |
| PS10 | 0.4 | 36.01 | 17 | 2364 |

### 3.2. Ultrasonic Pulse Velocity

The ultrasonic pulse velocity method is a non-destructive testing method widely used in concrete. It is less expensive and more efficient than destructive experiments because it does not need the destruction of concrete samples [28]. Due to the high sensitivity of the test, the UPV approach aids in detecting not only discontinuities in deep elements but also extremely small discontinuities [29].

In order to evaluate the quality of concrete after adding PSF, the UPV test was carried out on the control group and concrete with PSF additions of 0.2%, 0.4%, 0.6%, 0.8%, and 1.0%. The test procedure and results are listed in Figure 6a,b. As seen from the results, the wave velocity decreases with increasing PSF content, indicating that the porosity and inhomogeneity of the sample increase due to the inclusion of higher percentages of PSF, which may result in lower density. This is consistent with the results obtained from previous physical parameter tests. Abdulridha [30] also obtained a similar conclusion, they added waste rope fibers to concrete and found that UPV decreased gradually with the increase of fiber content. Using the correlation law between ultrasonic pulse velocity and concrete quality, the prediction model is established by the regression analysis method (Figure 6c).

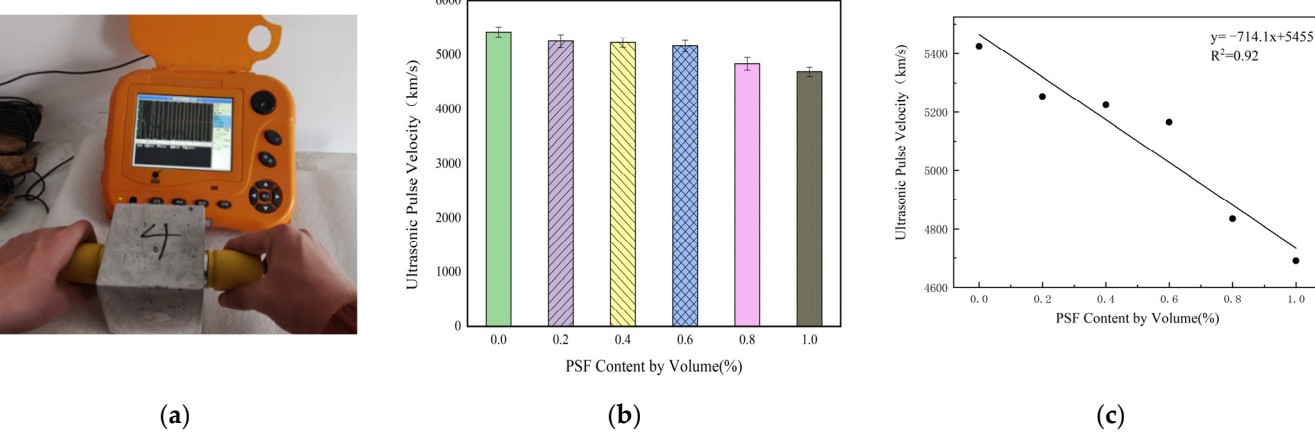

(a)                                (b)                                (c)

**Figure 6.** (**a**) UPV test process; (**b**) concrete UPV with different PSF content; (**c**) UPV results fitting regression.

### 3.3. Compressive Strength

For concrete materials, it is significant to obtain compressive strength data to evaluate their performance in engineering. Figure 7 and Table 5 summarize the results from the concrete compression test. It is obvious from the figure that the compressive strength of concrete generally shows a trend of first decreasing, then increasing, and then decreasing with the increase of PSF content. The PS6 had the greatest compressive strength, 7.3% higher than the control mix concrete. When compared to the control group, the PS4 and PS8 increased the strength by 0.2% and 1.6%, respectively, and correspondingly, the PS2 and PS10 also decreased the strength by 10.9% and 3.7%, respectively. The results from the data show that the inclusion of PSF had an evident effect on the compressive strength of concrete. When PSF were just added, due to few amounts, they may be unevenly distributed in the concrete, and the strength will decrease to a certain extent. Then, with the gradual increase of the volume content of PSF, the positive effect of fibers on the compressive strength of concrete began to appear. However, when a certain critical value is reached, continuing to increase the content will result in a decrease in strength. Similar results in compressive strength were obtained in the previous work of Akhmetov [31] and Orouji [32].

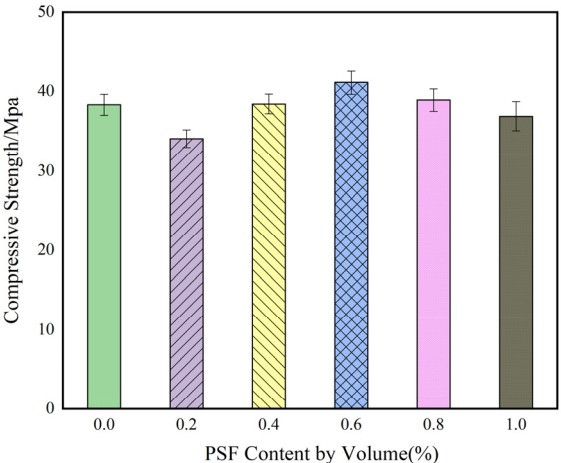

**Figure 7.** Concrete compressive strength with different PSF content.

**Table 5.** Results of mechanical experiments.

| Group Number | Compressive Strength/MPa | Splitting Tensile Strength/MPa |
|---|---|---|
| PS0 | 38.3 | 3.12 |
| PS2 | 34.1 | 3.36 |
| PS4 | 38.4 | 3.84 |
| PS6 | 41.1 | 4.04 |
| PS8 | 38.9 | 4.21 |
| PS10 | 36.9 | 4.48 |

On the one hand, the inclusion of PSF can reduce the generation of internal micro-cracks and effectively inhibit the development of existing cracks, preventing the occurrence of the unfavorable phenomenon that micro-cracks gradually develop into macro-cracks. On the other hand, PSF are scattered in the matrix and closely combined with the cement matrix to form a three-dimensional interlaced spatial network, which supports the aggregate particles and reduces the cracks caused by aggregate settlement. However, excessive PSF will lead to uneven distribution and aggregation of fibers, resulting in weakened interfacial area and decreased strength. This can be verified by a decrease in the strength of PS10 compared to the control group.

The failure morphology of PS0 and PS6 after reaching the maximum load is given in Figure 8. After reaching the maximum load, the pieces of PS6 remained on the concrete without appreciable detachment (Figure 8b). Correspondingly, the cement layer on the surface of PS0 had completely fallen off, and the concrete had lost the shape of the cube (Figure 8b). This phenomenon proves that PSF have a stronger bonding effect on the concrete fragments so that the concrete still maintains a relatively complete shape when it loses the bearing capacity, and the deformation resistance of the concrete is improved. After the concrete samples were completely broken, it could be observed that PSF still adhered to the broken pieces (Figure 8c), which demonstrated the linking roles of PSF at the macroscopic scale. The PSF bridge adjacent surfaces together and inhibit the creation and development of large cracks in concrete. The broken pieces in PS0 are not connected to each other, which is why the surface of PS0 falls off in a large area when reaching the maximum load (Figure 8c). At the same time, compared with common filamentary polypropylene fibers, PSF recovered from protective suits show a better bonding effect because PSF can connect larger concrete pieces.

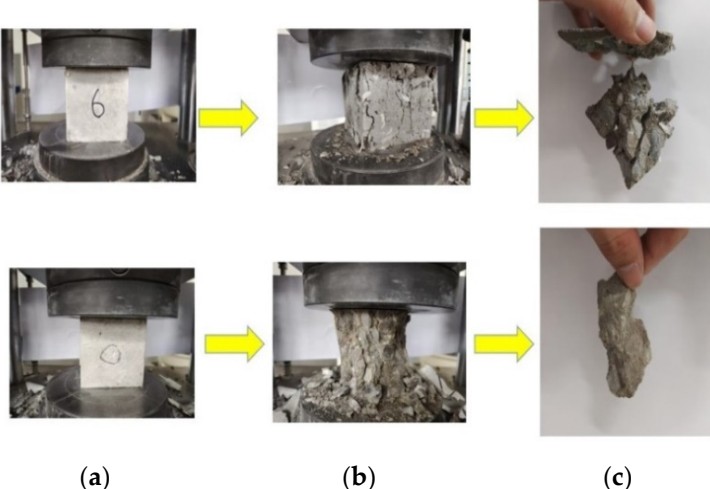

| (a) | (b) | (c) |

**Figure 8.** Failure behavior of PS0 and PS6 at maximum load: (**a**) before failure; (**b**) shows the final damaged specimens; (**c**) shows the pieces of each specimen.

*3.4. Splitting Tensile Strength*

The splitting tensile test results can be seen in Figure 9a and Table 5. The control mix showed a 28-day splitting tensile strength of 3.12 MPa. The splitting tensile strength of concrete steadily increases as the PSF content increases. The maximum tensile strength was reached at PS10, with a maximum value of 4.48 MPa, which was 43.6% higher than the control group. The PS2, PS4, PS6, and PS8 splitting tensile strengths were also enhanced by 7.7%, 23.1%, 29.5%, and 34.6%, respectively. It can be seen that PSF improve the splitting tensile strength of concrete in a complete manner, with the strength of each content improving when compared to the control group. In comparison to the minor increase in compressive strength, the addition of PSF resulted in a significant improvement in splitting tensile strength, suggesting that PSF contribute more to concrete splitting tensile strength than compressive strength. As seen in previous studies by Dharan [33], who found that adding 1% by volume of polypropylene fibers to concrete increased the tensile strength by 16.4%. Małek [34] and Mohammadhosseini [35] also observed a similar trend when incorporating recycled plastic fibers into concrete.

The PSF have the function of transmitting force and absorbing energy when concrete is under tension. By transmitting the load to other contact surfaces, they avoid stress concentration and improve the tensile performance of concrete. The improvement of the tensile strength of the experimental concrete also benefited from the anti-cracking effect of PSF. Since the tensile strength and ductility of PSF are higher than those of the cement matrix, when the macroscopic cracks propagate in the concrete, the fibers bridge across the cracks, which increases the crack propagation resistance and improves the fracture energy of the concrete. During concrete curing, concrete may develop micro-cracks due to internal autogenous and chemical shrinkage, PSF fill the distance between these micro-cracks, reducing the number of cracks and making the crack size smaller, thereby reducing the stress intensity factor at the crack, and the stress concentration at the tip is relieved. A regression prediction model for concrete splitting tensile strength and PSF volume content was established, as shown in Figure 9b.

The morphology of concrete after the splitting tensile test is given in Figure 10. The bridging phenomenon of the PSF connection cracks can be observed in Figure 10a. After the concrete is cracked, since the PSF bridge on both sides of the crack, the PSF concrete will not be brittle like the matrix but will be damaged due to fiber pulling. The fractured PSF were observed on the failure section of the concrete (see Figure 10b), which proved that the PSF was also loaded and pulled apart during the tensile process of the concrete. The PSF bear the stress transferred to them due to the cracking of the matrix, and the involvement of the fibers makes the matrix crack continuously and can further bear the load.

The concrete tensile–compression ratio is the ratio of splitting tensile strength to compressive strength, and it is an essential indicator of concrete brittleness. When the ratio of tensile–compression is larger, the brittleness of concrete will decrease, and the toughness will increase accordingly. Figure 9c shows the calculation results of the tension–compression ratio of each group. The results of the figure indicated that with the gradual increase of PSF content, the concrete tension–compression ratio shows an upward trend, which is 20.6–49.1% higher than that of the control group. The inclusion of PSF makes the concrete gradually transform from a brittle material to a plastic material. Similar conclusions can be acquired by observing the failure modes of PSF concrete under compression and tension.

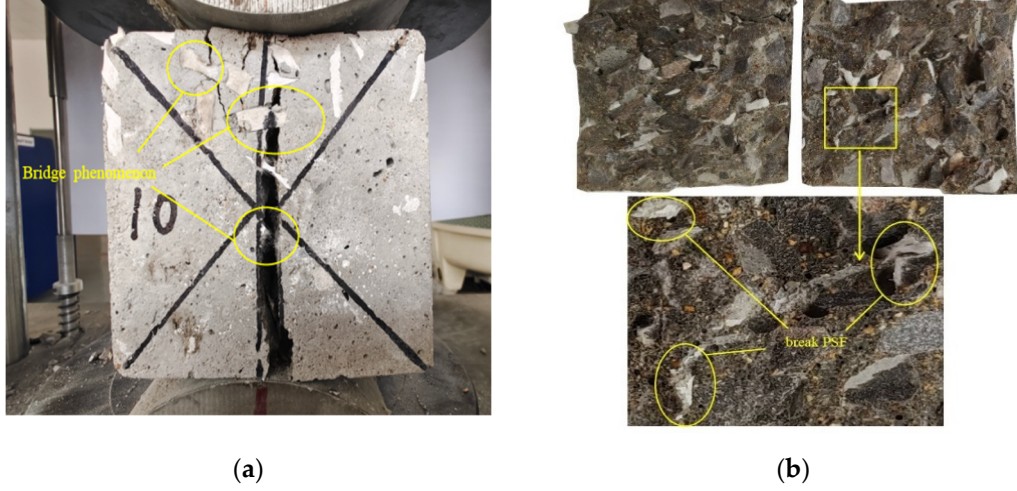

**Figure 9.** (**a**) Concrete splitting tensile strength with different PSF content. (**b**) Split tensile results fitting regression. (**c**) Concrete tensile–compression ratio.

**Figure 10.** Cracks of concrete after splitting test: (**a**) show PSF bridge phenomenon; (**b**) show concrete failure section.

### 3.5. Microscopic Electron Microscope Analysis

To study the effect of PSF on concrete microstructure, micrographs of the broken pieces of concrete shown in Figure 11 were obtained with a scanning electron microscope (SEM). It can be visible in a figure the rebar-like PSF are randomly distributed in the concrete and have good contact with the cement matrix (Figure 11a,b), which reduce the generation of micro-cracks in the concrete. The PSF exist between adjacent cement matrices, connecting cracks and transferring stress (Figure 11c), reflecting the bridging effect of PSF in the microscopic domain. It can be observed from Figure 11d that there are tiny voids between PSF, about a few microns, and the cement particles can enter and fill these voids, resulting in a tighter connection between the PSF and the entire cement matrix. The PSF are encapsulated by the cement matrix (Figure 11e,f). When the force acts on the microstructure, the fiber and the matrix bear the load together. Because of the tight connection between the PSF and the matrix, the fibers will not be pulled out but will continue to bear the load until they are broken. According to the findings of a similar study by Peled [36], fabric polypropylene fibers had a better binding ability to cement than regular polypropylene fibers. Simultaneously, these fibers can join cement and aggregates, increasing the bond between them and the microstructure of concrete.

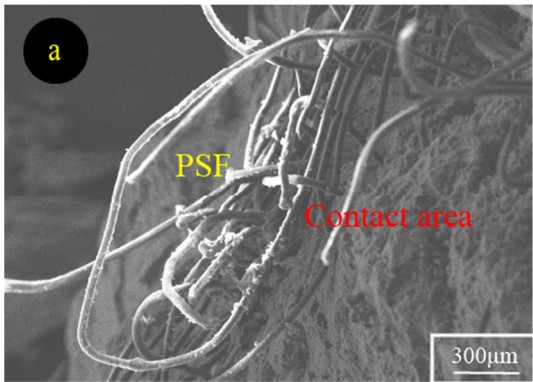
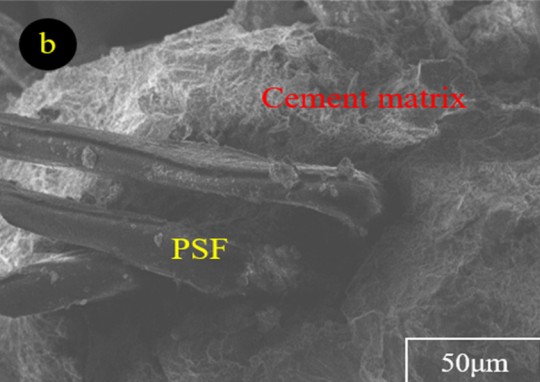

**Figure 11.** *Cont.*

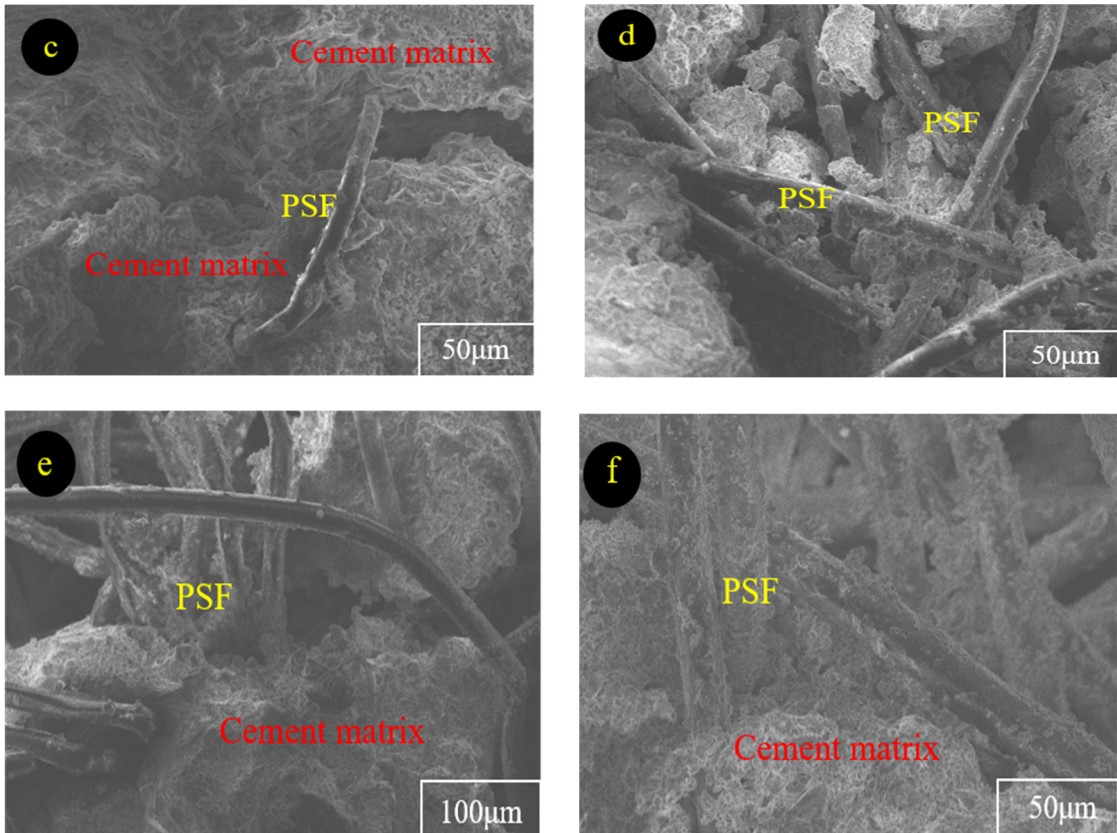

**Figure 11.** Microstructure of concrete with PSF: (**a**,**b**) PSF contact cement matrix; (**c**) PSF micro bridging; (**d**) cement particle filling phenomenon; (**e**,**f**) PSF is wrapped by cement matrix.

## 4. Finite Element Simulation

Polypropylene fibers are often combined with cement-based materials to be used as shotcrete in the construction of underground structures. Jiang [37] has studied the application of polypropylene fiber shotcrete in tunnel excavation engineering by means of numerical simulation and other methods. In order to explore the possibility of the practical application of PSF, the application of PSF shotcrete in roadway lining support was simulated using the finite difference method. The model is a semicircular arched structure with a radius of 2.5 m, a radial depth of 5 m, and a lining thickness of 200 mm. The axisymmetric model was chosen and used because it simulates lining structures in underground works and simplifies calculations. Since it is an axisymmetric graph, only half of it is simulated to reduce the amount of calculation. For comparison, ordinary concrete (concrete without added PSF) was also simulated. Figure 12 shows the geometry of the model. To reduce errors, the effect of mesh size is reduced by using fine meshing, as shown in Figure 13.

Table 6 lists the material properties of PSF shotcrete and ordinary shotcrete. Each material property is based on laboratory results and data reported in the literature. A uniform surface load of about 4 MPa was used to simulate the loading of the rock mass on the lining structure.

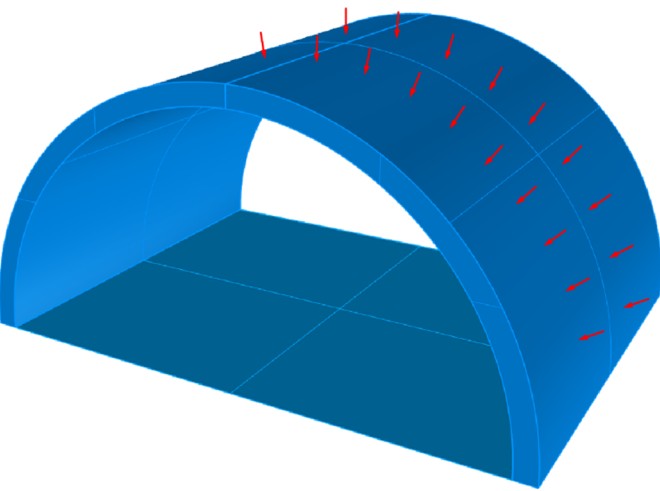

**Figure 12.** Finite element model geometry.

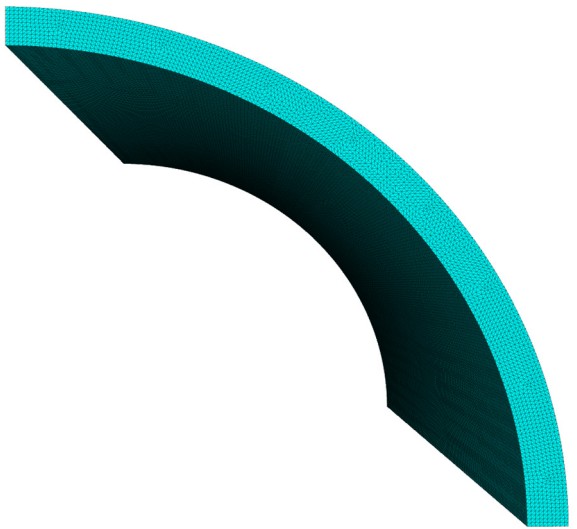

**Figure 13.** Model generated mesh.

**Table 6.** Materials properties.

| Properties | PSF Concrete | Ordinary Concrete |
| --- | --- | --- |
| Modulus of Elasticity (GN/m$^2$) | 28 | 30 |
| Poisson's ratio | 0.2 | 0.2 |
| C (kN/m$^2$) | 6780 | 5440 |
| $\phi$ (degree) | 53.42 | 58.02 |

Through the finite element simulation, the displacement, and stress of the structure can be accurately analyzed. Figures 14 and 15 show the vertical displacement of the PSF shotcrete lining structure and the ordinary shotcrete lining structure, respectively. It can be seen from the figure that when using PSF shotcrete as the lining structure, the vertical displacement of the structure increases slightly, from 5.3 mm to 5.8 mm, and the increase rate is about 9.4%. However, referring to the previous research, the tensile strength of PSF concrete is increased by 29.5%, which allows the PSF shotcrete lining structure to have greater deformation, and the support form is changed from rigid support to semi-rigid support, which exerts the bearing capacity of the rock mass itself. Therefore, PSF has good application value in the lining structure of underground engineering, which is of

great significance to improving the performance of the lining structure and the reuse of medical waste.

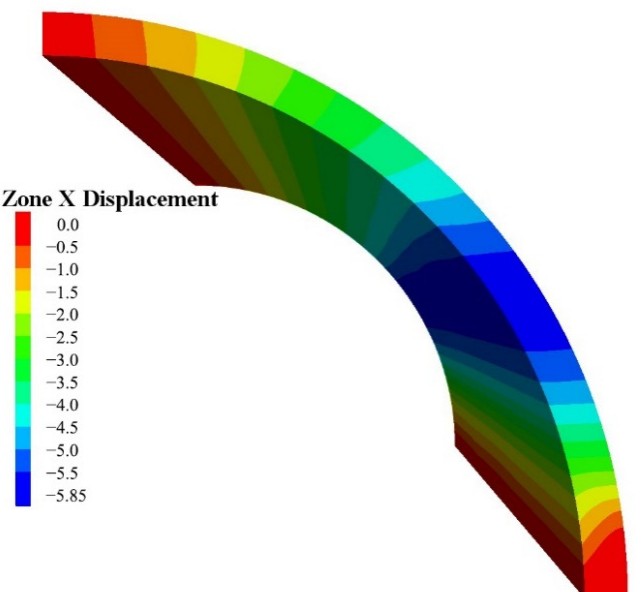

**Figure 14.** Vertical displacement of PSF shotcrete lining structure.

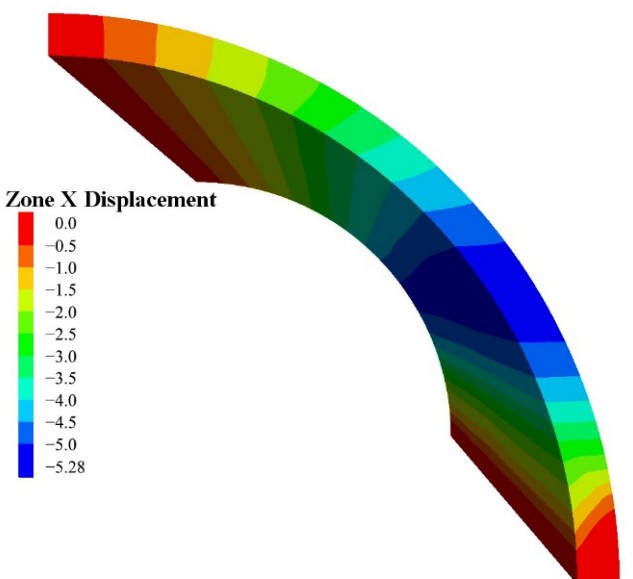

**Figure 15.** Vertical displacement of ordinary shotcrete lining structure.

## 5. Treatment Methods and Cost Analysis of PS

As the pandemic intensifies, global PPE waste management is facing challenges. Before the outbreak, the disposal of these wastes was a complex project. The primary approach to this dilemma in developing countries is landfilling [38]. However, this is clearly not a secure disposal strategy. In some larger economies, they have more rational solutions, such as China's deployment of mobile devices throughout Wuhan to collect and incinerate used medical waste [39]. While incineration is a common practice for disposing of plastic waste, it has a number of severe environmental consequences. The combustion of these PPE produces acid gases and aromatic compounds and the ash after combustion contains excessive heavy metals, which may pollute the atmosphere and water sources [40].

It will be an effective treatment method to use PPE and PS in engineering and concrete construction, but medical waste can contain bacteria and viruses, so it is necessary to thor-

oughly disinfect PPE before use to cut off the possibility of virus transmission. At present, the commonly used disinfection methods include high-temperature disinfection, chemical disinfection, and ultraviolet irradiation disinfection [41]. Li [42] reported an example of using high temperature to quickly purify N95 masks. Treatment at 100 °C for 30 s can kill most bacteria and viruses on the mask. Kampf [43] recommended a surface disinfection procedure using 62–71% ethanol, which can effectively inactivate the coronavirus present on plastic and glass. In addition, germicidal irradiation by ultraviolet light can also be used to kill viruses from masks [44,45]. At the same time, in order to reduce the risk of construction and transportation personnel being exposed to the virus, the machinery for transport and construction also needs to be supported by hygienic technology.

To simplify processing procedures and reduce operating costs, it is recommended to use high-temperature pretreatment to kill viruses when sterilizing PPE and PS on a large scale. Recycled PSF can be added to concrete to improve its mechanical properties, which has a similar effect as purchased polypropylene fibers. According to previous studies, the maximum volume parameter of PSF in concrete is 1%. Taking into account the disinfection cost of PS, collection, and cutting, and other factors, the total cost of PSF is 2458 ¥/ton (obtained by Equation (1), which is much lower than that of similar polypropylene fibers on the market (such as commercial-grade PPF 10,000 ¥/ton). From the economic cost perspective, the method proposed in this study to enhance concrete performance with PSF is competitive. Meanwhile, the cost is discussed preliminarily.

$$P = P_1 + P_2 + P_3 + P_4 \tag{1}$$

where $P$ is the total cost of obtaining the PSF, ¥/ton; $P_1$ is the transportation cost of the material, which is estimated to be 600 ¥/ton; $P_2$ is the cost of PS processing and cutting, which is estimated to be 800 ¥/ton; $P_3$ is the labor cost, which is estimated to be 1000 ¥/ton; $P_4$ is the cost of material disinfection, which is estimated to be 58 ¥/ton. $P_4$ is obtained by using Equations (2) and (3).

$$P_4 = P_e \times Q \tag{2}$$

$$Q \times \eta = c \times m \times \Delta T \tag{3}$$

$P_e$ is the price of industrial electricity in China, 1.025 ¥/kW h; $Q$ is the energy required to heat the recovered PS to 100 °C; $\eta$ is the heating conversion efficiency estimated to be 70%, and $c$ is the specific heat capacity of PS, which is 1.9 J/g °C; $m$ is the mass of the material to be processed; $\Delta T$ is the change in temperature during the heating process. According to the calculation, from room temperature (25 °C) to 100 °C, the energy required for disinfection treatment is 145,000 kJ/ton. Based on the heating conversion rate of 70%, the heating disinfection cost is 58 ¥/ton. This cost estimate is based on China's current electricity price standards and has a certain reference value.

## 6. Conclusions

This paper conducted a series of experiments to investigate the effect of adding recycled protective suit fibers (PSF) on the mechanical properties, microstructure, and workability of concrete. This study provides a new possibility to solve the medical waste generated by COVID-19. From the results obtained in this study, it can be concluded that:

1.  The inclusion of PSF improved the compressive and tensile strength of concrete, and the tensile strength increased by 43.6% compared with the control mix. The mechanical properties of concrete containing PSF have been significantly improved because of the influence of PSF in transferring stress, absorbing energy, and confining behavior on cracks.
2.  With the gradual increase of the PSF content, the tensile–compression ratio of concrete showed an upward trend, which was 20.6–49.1% higher than the control group. It indicated that the toughness of this concrete has been improved obviously.

3.  According to the SEM images, PSF had good contact with the cement matrix and hindered the propagation of micro-cracks. The introduction of PSF leads to the enhancement of the overall microstructure of concrete.
4.  The finite element simulation results prove that PSF shotcrete has good use value in the construction of underground lining structures.
5.  Applying COVID-19 protective suits to the production of high-quality concrete has the potential to show great environmental and economic benefits. The findings of this paper may help in the management of COVID-19 medical waste.

**Author Contributions:** Data curation, T.R.; Formal analysis, J.Z.; Funding acquisition, J.P.; Project administration, J.P.; Writing—original draft, T.R.; Writing—review & editing, J.Z. All authors have read and agreed to the published version of the manuscript.

**Funding:** This research received no external funding.

**Data Availability Statement:** The data used to support the findings of this study are available from the corresponding author upon request.

**Conflicts of Interest:** The authors declare that there are no conflicts of interest regarding the publication of this paper.

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
