# Peer review of "An Emerging Solution for Medical Waste: Reuse of COVID-19 Protective Suit in Concrete"

_sustainability, doi:10.3390/su141610045_

Round 1

Reviewer 1 Report

A lot of editing errors were listed in the manuscript,  please check and correct. The specific information was uploaded in the attachment.

Author Response

Response to Reviewer #1 Comments:

We are very grateful to your comments for the manuscript. According to your advice, we amended the relevant part in manuscript. All of your questions were answered one-by-one.

Point 1: The abstract should include the problem, objectives, method, results, conclusion, and recommendations. Please refer to the existing literature and improve the abstract.

Response 1: Reference is made to the existing literature Preliminary Evaluation of the Feasibility of Using Polypropylene Fibres from COVID-19 Single-Use Face Masks to Improve the Mechanical Properties of Concrete. J. Clean. Prod. 2021, 296, doi:10.1016/j.jclepro.2021.126460. The abstract has been refined and some discussion has been added, as detailed in the revised manuscript.

Point 2: Some keywords were used inappropriately. Keywords in paper should be the crucial words and repeated more in the whole paper. However, (Medical waste management, Protective suits) did not repeat in the main content.

Response 2: The keywords were amended by deleting the keyword "Medical waste management", adding the keyword "COVID-19", and amending the keyword "Protective suits" to "Protective suit fibers". Protective suits" was amended to "Protective suit fibers".

Point 3: Please note that the citation format should be consistent (Line 23, 25,45 ..., the same in the Section Introduction).

Response 3: The citation format for references has been standardized in the manuscript, and I apologize for the formatting errors that occurred.

Point 4: Which specification was the “P.C 42.5” stipulated in? Please add some details to the manuscript

Response 4: The P.C42.5 cement used in this experiment conforms to the Chinese standard GB175-2007 for ordinary silicate cement, which has a cured 28d compressive strength of 47.7 MPa, and some details have been added to the manuscript line 118-120.

Point 5: Obvious editing errors is listing in the manuscript (Line 121, 136, 147 and so on). Did you proofread the manuscript carefully?   

Response 5: Editing errors have been corrected in the manuscript, and the use of incorrect editing software led to such errors. The author apologizes for those editing errors.

Point 6: Units should be corrected. Like, Mpa should be MPa.

Response 6: All units have been corrected and the unit Mpa has been corrected to MPa.

Point 7: Grammar and punctuation errors left in the manuscript. Please check and correct.  

Response 7: The manuscript has been checked for grammar and punctuation, and corrections have been made.

Point 8: The layout of the pictures and tables in the manuscript is so weird, please try to adjust it.

Response 8: The images and tables in the manuscript have been rearranged, see the revised manuscript for details.

Point 9: Where’s the section 3.5?

Response 9: Section 3.4 was merged with section 3.5 in the first draft, but the title was not updated due to an error and has now been updated in the manuscript.

Point 10: There is a scale error in Figure 5.

Response 10: The protective clothing fibers in Figure 5 have been resized and the scale of the image has been improved, as shown in line 236 of the manuscript.

Point 11: The results are not obvious only depending on the pictures of the test part. Please add more chart to support your conclusion. 

Response 11: The results of the concrete mechanical experiments have been added to the manuscript and are shown in Table 5.

Point 12: What software do you use for simulation? What are your boundary conditions? Constraint form? Please specify if you want to use this section. Moreover, the simulation section doesn't seem to help with the content of the article. Please consult more literature to improve or delete this section.

Response 12: Numerical simulations were carried out using the Finite Difference Method (FDM) with displacement full constraints on the bottom face of the curved liner, symmetric constraints on the middle symmetric face, vertical displacement constraints on the front and back, and face forces applied on the outer curved face. The authors added numerical simulations with the aim of understanding the use of PSF concrete as shotcrete in tunnels as a means of exploring the application of PSF in practical engineering. In the reference (Assessment of the reuse of Covid-19 healthy personal protective materials in enhancing geotechnical properties of Najran’s soil for road construction: Numerical and experimental study. J. Clean. Prod. 320, 128772. https://doi.org/10.1016/j.jclepro.2021.128772), the researchers also performed HPPM (healthy personal protective materials) enhanced base were numerically simulated, so the authors believe that the addition of numerical simulation content would help the study.

Point 13: Please improve your language. Furthermore, the author should show deeper investigation in conclusion instead of showing only “good use”.

Response 13: The author's research on the application of protective suit fibers in concrete in this paper is only preliminary, and we may continue our research in the direction of the control of protective clothing fibers on concrete crack development and control of concrete self-shrinkage in subsequent work.

Thanks for your generous comments.

Reviewer 2 Report

The authors proposed to use the polypropylene fibers from protective suits in concrete. Overall, the paper is well written, and the findings and approach are interesting. There are some points that must be addressed to improve the clarity of the manuscript:

-There are many references in different formats throughout the manuscript and some with an "error" legend.

-In the introduction, the use of recycled PP in other applications should be discussed. Also, the use or impacts of protective equipment. For instance: Yang, S., Cheng, Y., Liu, T. et al. Impact of waste of COVID-19 protective equipment on the environment, animals and human health: a review. Environ Chem Lett (2022). https://doi.org/10.1007/s10311-022-01462-5

-The authors mentioned that the mechanical properties of concrete were enhanced, any idea about chemical properties? Resistance to temperature?

-What is the difference between adding this polymer to adding other polymers to concrete?

-On page 4, the authors briefly describe the process. It would be nice to see a schematic representation of the process (similar to a process flow diagram)

-In Table 4, the water-cement ratio and sand ratio are the same for all the group numbers.  Thus, it would be better to just mention them in the text and not in the table.

-Please include details about the software used for the finite element simulation.

-In the cost analysis, the authors should mention that it is preliminary because of its simplicity. The equations (1-3) must be aligned.

-The conclusions section should be presented in different paragraphs and not as bullet points.

Author Response

Response to Reviewer #2 Comments:

We are very grateful to your comments for the manuscript. According to your advice, we amended the relevant part in manuscript. All of your questions were answered one-by-one.

Point 1: There are many references in different formats throughout the manuscript and some with an "error" legend.

Response 1: Reference formatting has been standardized in the manuscript, and the locations where erroneous legends appeared have been corrected; apologies for any formatting errors.

Point 2: In the introduction, the use of recycled PP in other applications should be discussed. Also, the use or impacts of protective equipment. For instance: Yang, S., Cheng, Y., Liu, T. et al. Impact of waste of COVID-19 protective equipment on the environment, animals and human health: a review. Environ Chem Lett (2022). https://doi.org/10.1007/s10311-022-01462-5

Response 2: The reference has already been cited in the introductory section, see specifically lines 48-50 of the manuscript.

Point 3: The authors mentioned that the mechanical properties of concrete were enhanced, any idea about chemical properties? Resistance to temperature?

Response 3: The inclusion of polypropylene fibers may improve the durability performance of concrete, which can withstand high temperatures is also a part of durability, as mentioned in the Tanyildizi study (https://doi.org/10.1016/j.matdes.2008.11.032). In our subsequent work, we will also focus on the effect of PSF on the durability performance of concrete.

Point 4: What is the difference between adding this polymer to adding other polymers to concrete?

Response 4: The introduction of PSF in concrete can improve the mechanical properties of concrete, PSF is a type of polypropylene fiber, it has similar mechanical properties to other types of polypropylene fibers, but since PSF is obtained by cutting from protective clothing, he may not have the standard size of other polypropylene fibers.

Point 5: On page 4, the authors briefly describe the process. It would be nice to see a schematic representation of the process (similar to a process flow diagram)

Response 5: The authors thought that including a process flow diagram in the manuscript would help the reader understand the exact steps of the experiment, and thank you for your approval of the graphic.

Point 6: In Table 4, the water-cement ratio and sand ratio are the same for all the group numbers. Thus, it would be better to just mention them in the text and not in the table.

Response 6: In line 202 of the manuscript, it is mentioned that the water-cement ratio and sand rate are the same for all groups, and the same water-cement ratio and sand rate are mentioned again in the graph to draw the reader's attention to that property, which in turn reflects the effect of PSF on the workability of the concrete.

Point 7: Please include details about the software used for the finite element simulation.

Response 7: Simulation software developed based on the Finite Difference Method (FDM) was used in the numerical simulations. The constraints were set as displacement full constraints on the bottom surface of the curved liner, symmetry constraints on the middle symmetry surface, vertical displacement constraints on the front and back, and face forces applied on the outer curved surface. The authors concluded that the use of numerical simulation methods helped the study.

Point 8: In the cost analysis, the authors should mention that it is preliminary because of its simplicity. The equations (1-3) must be aligned.

Response 8: It has been described in the cost analysis, as specified in line 429 of the manuscript. Formatting errors in the formula have also been corrected.

Point 9: The conclusions section should be presented in different paragraphs and not as bullet points.

Response 9: Some changes have been made to the conclusions section, as described in lines 459-461 and 465-466 of the manuscript.

Thanks for your generous comments.
